# Erucic Acid-Rich Yellow Mustard Oil Improves Insulin Resistance in KK-A^y^ Mice

**DOI:** 10.3390/molecules26030546

**Published:** 2021-01-21

**Authors:** Asako Takahashi, Mayu Ishizaki, Yoshifumi Kimira, Yukari Egashira, Shizuka Hirai

**Affiliations:** 1Laboratory of Food Nutrition, Division of Applied Biochemistry, Graduate School of Horticulture, Chiba University, 648, Matsudo, Matsudo-shi, Chiba 271-8510, Japan; acaa1227@chiba-u.jp (A.T.); afga8471@chiba-u.jp (M.I.); egashira@faculty.chiba-u.jp (Y.E.); 2Faculty of Pharmacy and Pharmaceutical Sciences, Josai University, 1-1 Keyakidai, Sakado-shi, Saitama 350-0295, Japan; kimira@josai.ac.jp

**Keywords:** yellow mustard oil, erucic acid, obesity, type 2 diabetes, insulin resistance, inflammation

## Abstract

Obesity is a major risk factor for some metabolic disorders including type 2 diabetes. Enhancement of peroxisome proliferator-activated receptor (PPAR) γ, a master regulator of adipocyte differentiation, is known to increase insulin-sensitive small adipocytes. In contrast, decreased PPARγ activity is also reported to improve insulin resistance. We have previously identified erucic acid as a novel natural component suppressing PPARγ transcriptional activity. In this study, we investigated the effect of erucic acid-rich yellow mustard oil (YMO) on obese/diabetic KK-A^y^ mice. An in vitro luciferase reporter assay and mesenchymal stem cell (MSC) differentiation assay revealed that 25 µg/mL YMO significantly inhibited PPARγ transcriptional activity and differentiation of MSCs into adipocytes but promoted their differentiation into osteoblasts. In KK-A^y^ mice, dietary intake of 7.0% (*w*/*w*) YMO significantly decreased the surrogate indexes for insulin resistance and the infiltration of macrophages into adipose tissue. Furthermore, 7.0% YMO increased bone mineral density. These results suggest that YMO can ameliorate obesity-induced metabolic disorders.

## 1. Introduction

Obesity is a major risk factor for atherosclerosis, insulin resistance, type 2 diabetes, cardiovascular disease, hypertension, and dyslipidemia, called metabolic syndrome [1]. Obesity is considered to be the result of an increase in the number and size of adipocytes [2]. Peroxisome proliferator-activated receptor (PPAR) γ, the master regulator of adipocyte differentiation, is a ligand-responsive transcription factor. Synthetic PPARγ agonists such as thiazolidinediones are used as antidiabetic agents because they strongly promote adipocyte differentiation and increases insulin-sensitive, small adipocytes [3,4]. However, such PPARγ agonists are associated with some side effects, including body weight gain [5] and bone loss [6,7]. On the other hand, it has been reported that high-fat diet (HFD)-induced adipocyte hypertrophy and insulin resistance are improved in heterozygous PPARγ-deficient mice as compared with the wild type mice [8,9,10]. Synthetic PPARγ antagonists, GW9662 and bisphenol A diglycidyl ether, have also been reported to improve insulin resistance and obesity [11,12]. Furthermore, these PPARγ antagonists increase bone mass and osteoblast differentiation from mesenchymal stem cells (MSCs) [13,14]. Therefore, moderately decreased PPARγ activity is expected to improve insulin resistance and increase bone mass without inducing obesity.

In our previous study, we found that erucic acid isolated from rosemary promotes the differentiation of MSCs into osteoblasts but inhibits the differentiation into adipocytes by suppressing PPARγ transcriptional activity [15]. Erucic acid is one of the monounsaturated fatty acids widely found in Brassicaceae plants such as rapeseed and mustard [16,17]. In particular, mustard oil is rich in erucic acid [18]. However, there are few reports on the biological function of erucic acid.

In the present study, we investigated the effect of yellow mustard oil (YMO), which contains a large amount of erucic acid (approximately 37.0% in fatty acids), on metabolic disorders in obese/diabetic KK-A^y^ mice. We revealed that 25 µg/mL YMO significantly decreased PPARγ transcriptional activity and inhibited MSCs differentiation into adipocytes but enhanced their differentiation into osteoblasts. In KK-A^y^ mice, 7.0% YMO intake decreased surrogate indexes for insulin resistance and improved inflammation in adipose tissue. Furthermore, 7.0% YMO increased bone mineral density. These findings indicate that YMO may have beneficial effects on improving glucose and bone metabolism disorders induced by obesity and diabetes.

## 2. Results

### 2.1. YMO Suppressed PPARγ Transcriptional Activity and Differentiation of MSCs into Adipocytes

Since most fatty acids in vegetable oils are bound to triglyceride [19], we used the YMO hydrolyzed to free fatty acids in in vitro experiments. When we examined the effect of hydrolyzed YMO on PPARγ transcriptional activity by luciferase reporter assay, 25 μg/mL YMO and 5 μM GW9662 significantly decreased troglitazone-induced PPARγ transcriptional activity (Figure 1A). YMO used in this study contained 36.97% erucic acid (Figure 2, Table 1). The hydrolyzed YMO also showed a similar fatty acid composition (Appendix A). Therefore, YMO that showed significant decrease of PPARγ transcriptional activity contained about 27.24 μM erucic acid, which is comparable to the erucic acid concentration (25 μM) that suppressed PPARγ activity in our previous study [15]. Inhibitors of PPARγ transcriptional activity have been reported to regulate MSCs differentiation [14]. In the present study, 15–40 μg/mL YMO significantly decreased the number of Oil Red O^+^ adipocytes (Figure 1B) and increased that of alkaline phosphatase (ALP) ^+^ osteoblasts (Figure 1C).

To reveal the molecular mechanism of which YMO suppressed adipocyte differentiation and enhanced osteoblast differentiation from MSCs, we measured the expression of adipocyte and osteoblast differentiation-related genes by real time PCR. Twenty-five μg/mL YMO significantly decreased mRNA expression of PPARγ at day 12 and 14 (Figure 3A). The expression of PPARγ target genes, aP2 and LPL, were suppressed by YMO earlier than the decrease in PPARγ mRNA expression (Figure 3B,C). On the other hand, mRNA expression of osteoblast differentiation markers, ALP and Col1, were increased by YMO at days 12 and 14 (Figure 3F) and at days 8 and 12 (Figure 3G), respectively. The expression of Runx2 and Osterix, which are known as transcription factors involved in the induction of ALP and Col1 expression [20,21], were largely unaffected by YMO (Figure 3D,E).

### 2.2. YMO Suppressed Fat Accumulation in KK-A^y^ Mice

PPARγ antagonists have been reported to suppress HFD diet-induced obesity [11,12]. Therefore, we examined the effect of YMO on fat accumulation in KK-A^y^ mice. As a result, 3.5% YMO intake significantly reduced body weight and subcutaneous adipose tissue weight (Table 2). On the other hand, perirenal adipose tissue weight was significantly reduced in 7.0% YMO group (Table 2), which was accompanied by decreased mRNA expression of PPARγ target genes, *aP2* and *LPL* (Figure 4). However, GW9662 did not affect either body weight, adipose tissue weight, or expression of adipocyte differentiation marker genes (Table 2, Figure 4).

### 2.3. YMO Improved Insulin Resistance in KK-A^y^ Mice

To examine the effect of YMO on obesity-related development of insulin-resistance, we performed oral glucose tolerance test (OGTT). GW9662 significantly decreased plasma glucose and insulin levels compared to the control group (Figure 5A–D), leading to the significant decrease of homeostasis model assessment-insulin resistance (HOMA-IR) (Figure 5E) and increase of quantitative insulin-sensitivity check index (QUICKI) (Figure 5F) values, which are the surrogate indexes for insulin resistance. Although YMO did not significantly affect the area under the curve (AUC) of plasma glucose level (Figure 5B), AUC of plasma insulin level in 3.5% and 7.0% YMO groups was significantly decreased compared to the control group (Figure 5D). HOMA-IR value in 7.0% YMO group and QUICKI value in 3.5% and 7.0% YMO group were also significantly decreased (Figure 5E) and increased (Figure 5F), respectively, compared to the control group.

### 2.4. YMO Suppressed Infiltration of Macrophages into Adipose Tissue in KK-A^y^ Mice

We next investigated the morphological changes of perirenal adipose tissue in KK-A^y^ mice. Although YMO and GW9662 did not affect adipocyte size and number (Appendix A), 7.0% YMO and GW9662 significantly decreased crown-like structures (CLS) (Figure 6A,B) and F4/80^+^ (Figure 6A,C) areas. Furthermore, 3.5% and 7.0% YMO and GW9662 significantly decreased both the areas positive for a M1 macrophage marker, CD11c (Figure 6D), and a M2 macrophage marker, CD206 (Figure 6E). On the other hand, the ratio of CD206/CD11c area was significantly increased by 7.0% YMO and GW9662 groups (Figure 6F). Although mRNA expression levels of a macrophage marker, F4/80 (Figure 7A), and monocyte chemoattractant protein-1 (MCP-1) (Figure 7B), involved in macrophage infiltration, were not significantly different among the groups, mRNA expression of M1 macrophage markers, CD11c, Interleukin-1β (IL-1β), and tumor necrosis factor-α (TNF-α) (Figure 7C–E), and a M2 macrophage marker, CD206 (Figure 7F), in perirenal adipose tissue were significantly decreased in 3.5 and/or 7.0% YMO and GW9662 groups. Moreover, 7.0% YMO significantly decreased mRNA expression of another M2 macrophage marker, Ym1 (Figure 7G), but did not affect mRNA expression of arginase 1 (Arg1) (Figure 7H).

### 2.5. YMO Increased Bone Mineral Density (BMD) and Osteoblastic Bone Formation Marker

It has been reported that bone mineral density (BMD) decreases in KK-A^y^ mice as the progression of obesity [22]. In the present study, total, cortical, and cancellous BMD in the right femur was significantly increased in 7.0% YMO group (Figure 8A–C). Similar result was observed in left femur (Appendix A). Furthermore, plasma ALP activity, a marker of osteoblastic bone formation, was increased in 7.0% YMO group (Figure 8D), whereas plasma tartrate-resistant acid phosphatase (TRAP) activity, a marker of osteoclastic bone resorption, was not affected by YMO and GW9662 (Figure 8E).

## 3. Discussion

Adipose tissue plays an important role in the regulation of glucose metabolism throughout the body. It is reported that many kinds of synthetic [23,24] and natural [25,26] PPARγ agonists increase the number of insulin-sensitive small adipocytes by promoting adipocyte differentiation. On the other hand, it has been reported that heterozygous knockout of PPARγ or moderate suppression of PPARγ activity also contribute to enhancement of insulin sensitivity without increasing adipocyte mass [8,9,10]. However, unlike PPARγ agonists, there are few natural components that have been reported to inhibit PPARγ transcriptional activity [27,28].

We previously identified erucic acid as a novel natural component with inhibitory effect of PPARγ transcriptional activity [15]. To evaluate the in vivo function of erucic acid, we used YMO, which is rich in erucic acid, in the present study. Twenty-five μg/mL YMO (corresponding to 27.24 μM erucic acid) significantly decreased PPARγ reporter activity and enhanced MSCs differentiation into osteoblasts rather than adipocytes. YMO dose-dependently decreased PPARγ reporter activity at a concentration of 20 or 25 μg/mL or less but restored the activity at concentration above 25 μg/mL. A similar result was shown with the effect of YMO on MSCs differentiation. YMO dose-dependently suppressed adipocyte differentiation and enhanced osteoblast differentiation at concentrations of 30 μg/mL or less, but the effect was attenuated at concentration above 30 μg/mL. In our previous study using erucic acid, the effects of erucic acid on decreasing PPARγ activity and regulating MSCs differentiation were also attenuated at concentrations above 25 μM [15], which may be due to changes in solubility with morphological changes of fatty acid depending on its concentration [29]. Further study will be needed to clarify the relationship between concentration-related morphological changes of erucic acid in YMO and its effect on PPARγ activity and differentiation of MSCs.

When we examined the in vivo effect of YMO in a KK-A^y^ obese/insulin-resistant model mice, YMO reduced perirenal adipose tissue weight and mRNA expression of PPARγ target genes but did not affect PPARγ mRNA expression. PPARγ agonists are known to increase the number of small adipocytes by promoting adipocyte differentiation [3,4], but YMO did not affect the average area and number of adipocytes. Furthermore, YMO also reduced the expression of PPARγ target genes earlier than the reduction of PPARγ mRNA expression in MSCs. These results suggest that YMO suppresses adipocyte differentiation by reducing PPARγ activity rather than by reducing its mRNA expression. On the other hand, GW9662 showed no effect on body weight and adipose tissue weights. Previous report has shown that 0.1% GW9662 suppresses HFD-induced fat accumulation and weight gain in C57BL/6J mice [11]. Therefore, it was suggested that GW9662 concentration used in this study was insufficient to suppress fat accumulation in KK-A^y^ mice.

Obesity-induced insulin resistance is closely associated with macrophage infiltration into adipose tissue [30]. In obese animals, macrophages infiltrate into adipose tissues recruited by MCP-1 secreted from hypertrophied adipocytes [31]. Excess MCP-1 and inflammatory cytokines including TNF-α secreted from infiltrated macrophages, induce additional infiltration of macrophages and insulin resistance, respectively [32]. It is also known that the phenotype of macrophages in adipose tissue changes as the progression of obesity. M1 macrophages which increase with the progression of HFD-induced obesity highly express pro-inflammatory cytokines, including TNF-α, which is considered to be involved in insulin resistance [33]. On the other hand, M2 macrophages expressing anti-inflammatory cytokines are abundant in the adipose tissues in lean animals [34]. In this study, YMO and GW9662 improved HFD-induced insulin resistance. YMO and GW9662 decreased the area of CLS, an accumulation of immune cells surrounding adipocytes [35], and the positive area of F4/80, a macrophage marker expressed in both M1 and M2 [33] in perirenal adipose tissue. YMO and GW9662 reduced positive areas for both a M1 macrophage marker, CD11c, and a M2 macrophage marker, CD206. The mRNA expressions of M1 macrophage markers (*CD11c*, *IL-1β*, *TNF-α*) and M2 macrophage markers (*CD206*, *Ym1*) were also decreased by YMO and GW9662. However, M2/M1 ratio in adipose tissue was significantly increased by YMO and GW9662. These results suggest that YMO and GW9662 may increase insulin sensitivity by suppressing macrophage infiltration and increasing M2/M1 ratio in adipose tissue. Several reports have shown that the insulin-sensitizing effect of PPARγ ligands are exerted not by classical agonism but by inhibition of cyclin-dependent kinase (CDK) 5-mediated phosphorylation of PPARγ at Ser^273^ (pS273) [36,37,38]. PPARγ agonists are also known to exert anti-inflammatory effects through suppression of nuclear factor-kappa B (NF-κB) transcriptional activity without classical agonism [39]. Furthermore, PPARγ antagonist Gleevec is also suggested to ameliorate adipose tissue inflammation via inhibition of pS273 and NF-κB activation [40]. Therefore, YMO and GW9662 may suppress adipose tissue inflammation via similar mechanism. Further study is needed to clarify the detailed mechanism of anti-inflammation by YMO and GW9662.

In the present study, YMO promoted the differentiation of MSCs into osteoblasts, which was accompanied by increased mRNA expression of osteoblast differentiation markers, *ALP* and *Col1*, but not by the changes of mRNA expressions of *Runx2* and *Osterix*. Runx2 and Osterix are known as transcription factors involved in the induction of *ALP* and *Col1* expressions [20,21], but several studies have reported that *ALP* expression can be induced without changes of *Runx2* mRNA expression [41,42,43,44,45], which suggests the involvement of the binding activity of Runx2 to *ALP* promoter [41,42,43] or of other transcription factors [44,45]. In addition to obesity and diabetes, KK-A^y^ mice are reported to show a significantly lower BMD and blood level of bone formation markers compared to C57BL/6 normal mice, although they have no effect on blood bone resorption markers [22]. Therefore, we examined the effect of YMO on BMD and plasma bone metabolism markers in KK-A^y^ mice. In this study, 7.0% YMO significantly increased femoral BMD and plasma level of bone formation, indicated by ALP activity, without affecting the plasma level of bone resorption, indicated by TRAP. These results suggest that YMO may also suppress the decrease in BMD associated with diabetes by affecting osteoblast differentiation and/or its function. On the other hand, GW9662 did not affect BMD in KK-A^y^ mice. Beekman et al. [46] reported that administration of GW9662 did not affect bone loss in ovariectomized (OVX) mice. OVX-induced bone loss is a model of “high-turnover osteoporosis” often observed in postmenopausal women [47], caused by osteoclastic bone resorption greatly exceeding osteoblastic bone formation. On the other hand, the diabetic animal used in this study is a model of “low-turnover osteoporosis” in which both bone formation and resorption are lower than normal, but even lower bone formation than resorption results in bone loss [48]. Therefore, components that activate osteoblastic bone formation may be effective in improving bone loss in diabetic models. To clarify the mechanism of increased BMD by YMO, it is necessary to examine the effect of higher concentration of GW9662 and/or the difference between GW9662 and YMO uptake into bone.

## 4. Materials and Methods 

### 4.1. Chemicals

YMO was provided from Kewpie Corporation (Tokyo, Japan). A synthetic PPARγ agonist, troglitazone, and a synthetic PPARγ antagonist, GW9662, were purchased from FUJIFILM Wako Pure Chemical Corporation (Osaka, Japan). Ethanol and dimethyl sulfoxide (DMSO) were also obtained from FUJIFILM Wako Pure Chemical Corporation.

### 4.2. Hydrolyzation of YMO

For in vitro experiments, YMO was hydrolyzed to free fatty acids as follows; 100 mL of 1M KOH/ethanol solution was added to 10 g of silica gel-treated YMO and heated in a water bath at 80–100 °C for 30 min, and then 200 mL of water and 100 mL of diethyl ether were added and distributed. One hundred milliliter of diethyl ether and 100 mL of water were added to the aqueous layer and the ether layer, respectively, and further distributed. Approximately 6 mL of 4N HCl was added to the aqueous layer containing the saponified product, and then 100 mL of diethyl ether was added for further distribution. After removing water from the diethyl ether layer using sodium sulfate, a filtered and distilled sample was used for in vitro experiment.

### 4.3. Fatty Acid Analysis of YMO

Fifty milligrams of YMO or its hydrolyzed sample (each from Kewpie Corporation) and 5 mL of 0.5 mol/L sodium hydroxide-methanol solution were mixed and heated in a boiling water bath for 5–10 min. After adding 7 mL of boron trifluoride-methanol reagent (Sigma-Aldrich Co., St. Louis, MO, USA) and boiled for 2 min, 5 mL of hexane was added and heated for another 1 min. After adding about 140 mL of saturated saline, 1 mL of the hexane layer was transferred to a test tube and about 0.2 g of sodium sulfate was added to remove water, which was used as a sample solution for fatty acid analysis by gas chromatograph with flame ionization detector (GC-FID) (GC-6890N; Agilent Technologies Inc, Santa Clara, CA, USA) using an Omegawax^®^ 250 Intuvo Capillary GC Column (L × I.D. 30 m × 0.25 mm, df 0.25 μm; Sigma-Aldrich, Supelco, PA, USA). The detected fatty acids were quantified with Supelco^®^ 37 Component FAME Mix (Sigma-Aldrich, Supelco, PA, USA).

### 4.4. PPARγ Luciferase Reporter Assay

We performed a luciferase reporter assay according to our previous report [15]. Briefly, monkey CV-1 kidney cells (American Type Culture Collection, USA) reached 90% confluence on a 100-mm culture plate were transfected with a reporter vector (p4 × UASg-tk-Luc), an expression vector for a chimera protein for GAL4 DNA-binding domain and human PPARγ ligand-binding domain (pM-hPPARγ), and an internal control vector (pRL-CMV) for 4 h. These vectors were kindly provided by Prof. Teruo Kawada and Dr. Tsuyoshi Goto, Kyoto University. After transfection, cells were seeded at 5.00 × 10^3^ cells/well on 96 well plates and treated with medium containing 5 µM troglitazone (FUJIFILM Wako) dissolved in DMSO and either each concentration (0, 10, 20, 25, 30, 40 µg/mL) of YMO dissolved in ethanol or 5 µM GW9662 (FUJIFILM Wako) dissolved in ethanol for 24 h. Luciferase reporter assay was performed using Dual-Glo luciferase assay system (Promega, Madison, WI, USA).

### 4.5. Culture of C3H10T1/2 Cells

Mouse C3H10T1/2 mesenchymal stem cells (JCRB Cell Bank, Osaka, Japan) were cultured in DMEM containing 10% FBS (Equitech-Bio Inc, Kerrville, TX, USA) and 100 U/mL penicillin and 100 µg/mL streptomycin (FUJIFILM Wako) at 37 °C under a humidified 5% CO_2_ atmosphere. Cells were seeded at 1.25 × 10^5^ cells/well on 24 well plates for staining or 2.50 × 10^5^ cells/well on 12 well plates for RNA extraction. After reaching confluent, the culture medium was replaced with DMEM containing 10% FBS and 0, 15, 20, 25, 30, 35, or 40 µg/mL YMO dissolved in ethanol and changed every 2 days for 14 days.

### 4.6. Oil Red O and ALP Staining

Differentiated adipocytes and osteoblasts were evaluated by Oil Red O staining and ALP staining, respectively, according to the methods previously described [15]. Briefly, in Oil Red O staining, cells were stained with Oil Red O solution (0.5% Oil Red O/2-propanol diluted in water [3:2]) for 1 h. In ALP staining, cells were stained using Alkaline Phosphatase Staining kit (Cosmo Bio Co., LTD, Tokyo, Japan) according to the manufacturer’s protocol. After staining, a grid seal (1806-009, AGC TECHNO GLASS Co., Ltd., Shizuoka, Japan) was attached to the bottom of the plate, and the number of Oil Red O^+^ cells/well or that of ALP^+^ cells/well were counted under a light microscope.

### 4.7. Animal Experiments

Four-week-old male KK-A^y^ mice were purchased from CLEA Japan, Inc. (Tokyo, Japan). Mice were housed in individual cages, under a 12-h light/dark cycle and 22 ± 1 °C and were given free access to distilled water. Mice were divided into five groups and fed with one of the following diets for 16 weeks: control diet (a modified AIN-93G containing 10% (*w*/*w*) fat) (*n* = 11), control diet containing 0.005% (*w*/*w*) GW9662 (*n* = 10), 1.0% (*w*/*w*) YMO (*n* = 10), 3.5% (*w*/*w*) YMO (*n* = 10), or 7.0% (*w*/*w*) YMO (*n* = 10). The energy intake of each mice was adjusted by pair feeding.

After 16 weeks of respective dietary treatment, all mice were killed under anesthesia after 16 h of fasting, and blood, each tissue, and femur were collected. All animal experiments were carried out in accordance with the Ethical Guidelines for the Care and Use of Laboratory Animals, Chiba University. The present study was approved by the Ethics Committee for Animal Experiments of Chiba University (Approval No. 30-387).

### 4.8. OGTT

After 15 weeks of feeding, KK-A^y^ mice were fasted for 16 h and administered with glucose at 1.5 g/kg body weight by oral gavage. Approximately 40 µL of blood samples were collected from the tail vein at 0, 15, 30, 60, 120 min after glucose administration. Plasma glucose and insulin concentration were measured using Glucose CII-test Wako (FUJIFILM Wako) and Morinaga Ultra-Sensitive Mouse Insulin ELISA Kit (Morinaga Institute of Biological Science, Yokohama, Japan), respectively, according to each manufacturer’s protocol. As the index for estimating insulin resistance, homeostasis model assessment-insulin resistance (HOMA-IR) and quantitative insulin-sensitivity check index (QUICKI) were calculated from fasting plasma glucose and insulin concentrations as follows: HOMA-IR = (plasma glucose (mmol/L) at 0 min) × (plasma insulin (μU/mL) at 0 min)/22.5 [49], QUICKI = 1/(log (plasma insulin (μU/mL) at 0 min) + log (plasma glucose (mg/dL) at 0 min)) [50].

### 4.9. Quantitative Real-Time PCR Analysis

Total RNA was extracted from C3H10T1/2 cells after 14 days of culture and perirenal adipose tissue of KK-A^y^ mice using RNAiso Plus (Takara Bio inc., Shiga, Japan) according to the manufacturer’s protocol. cDNA was synthesized with ReverTra Ace^®^ qPCR RT Master Mix with gDNA Remover (TOYOBO CO., LTD., Osaka, Japan) according to the manufacturer’s protocol. Quantitative real-time PCR analysis was performed with Applied Biosystems^®^ StepOnePlus real-time PCR system (Thermo Fisher Scientific, Waltham, MT, USA) using THUNDERBIRD^®^ SYBR^®^ qPCR Mix (TOYOBO). The PCR amplification was performed as described previously [15]. The primer sequences are listed in Table 3. The expression level of *36B4* mRNA was used as the internal standard for the determination of each target mRNA expression levels.

### 4.10. Histological Analysis of Adipose Tissues

Perirenal adipose tissues were fixed with 10% formalin for overnight and stored in PBS at 4 °C. Each perirenal adipose tissue was embedded in paraffin and sliced at 0.5 µm using Leica SR 2000 Microtome (Leica Biosystems Nussloch GmbH, Wetzlar, Germany). For hematoxylin and eosin (HE) staining, deparaffinized sections were incubated with Mayer’s Hematoxylin Solution (FUJIFILM Wako) for 2 min and 0.25% Eosin Y (FUJIFILM Wako) for 5 min. For immunochemical staining, deparaffinized sections were blocked with 10% Normal Serum Block (SIG-31172, BioLegend, Inc., San Diego, CA, USA) and 2% BSA (Sigma-Aldrich Co.) solution and were incubated with an anti-mouse F4/80 (F4/80 antibody, Cl:A3-1, Bio-Rad Laboratories, Inc., Hercules, CA, USA) (1:200), anti-mouse CD11c (Anti CD11c/Integrin α X, 17342-1-AP, Proteintech Group, Inc., Rosemont, IL, USA) (1:500), or anti-CD206 (Anti-Mannose Receptor antibody, ab64693, Abcam, Cambridge, UK) (1:5000) as the primary antibody at 4 °C overnight. Antigen signals were detected by incubation with biotinylated anti-IgG antibody (Rabbit Anti-Rat IgG Antibody, BA-4001, Vector Laboratories, Burlingame, CA, USA, or Biotin-Conjugated Goat Anti-Rabbit IgG Secondary Antibody, bs-0295G-Biotin, Bioss Antibodies Inc., Woburn, MT, USA), followed by streptavidin-horseradish peroxidase (Abcam) and then DAB (Dojindo Molecular Technologies, Inc., Kumamoto, Japan). Each stained slide was observed under light microscopy and analyzed by Win-ROOF ver. 7.2 (MITANI Corporation, Tokyo, Japan).

### 4.11. Analysis of Bone Mineral Density (BMD)

The femur of each mouse was used for the analysis of BMD by peripheral quantitative computed tomography (pQCT: LaTheta LCT-100, ALOKA, Tokyo, Japan). BMD was calculated from the bone mineral content of the measured area.

### 4.12. Measurement of Plasma Bone Metabolic Markers

Plasma ALP activity, a marker of osteoblastic bone formation, and TRAP activity, a marker of osteoclastic bone resorption was measured by Labo assay^TM^ ALP (FUJIFILM Wako) and TRACP & ALP Assay Kit (Takara Bio inc.), respectively, according to each manufacturer’s protocol.

### 4.13. Statistical Analysis

Data are presented as mean ± standard error (SE). Statistical analysis was performed using Bell Curve Excel-Toukei (Social Survey Research Information Co., Tokyo, Japan). The normalization of data was evaluated by Shapiro–Wilk test. Reporter assay data were analyzed by Student’s t-test. The other data were analyzed using one-way analysis of value (ANOVA) followed by Dunnett’s test or Steel’s test. Differences were considered significant at *p* < 0.05.

## 5. Conclusions

We demonstrated that YMO, rich in erucic acid, improved adipose tissue inflammation and insulin resistance. It was also revealed that YMO contributes to the increase of BMD in diabetic mice through the enhancement of bone formation. These results suggest that YMO may contribute to the improvement of impaired glucose and bone metabolism associated with obesity and diabetes.

## Figures and Tables

**Figure 1 molecules-26-00546-f001:**
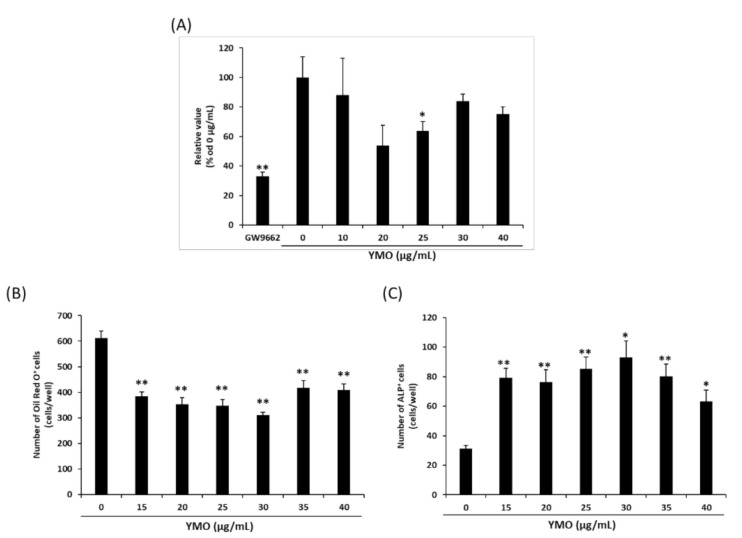
Effect of YMO on peroxisome proliferator activated receptor γ (PPARγ) transcriptional activity and differentiation of C3H10T1/2 cells. (**A**) Effect of 0–40 µg/mL YMO on PPARγ transcriptional activity in the presence of 5 µM troglitazone. Relative luciferase activities were expressed as percentage of vehicle control. (**B**,**C**) Effect of 0–40 µg/mL YMO treatment for 14 days on differentiation of C3H10T1/2 cells. A number of Oil Red O^+^ cells (**B**) and of ALP^+^ cells (**C**) in each well were counted under light microscope. Values are mean ± SE, *n* = 4–5. ** *p* < 0.01, * *p* < 0.05 versus 0 µg/mL.

**Figure 2 molecules-26-00546-f002:**
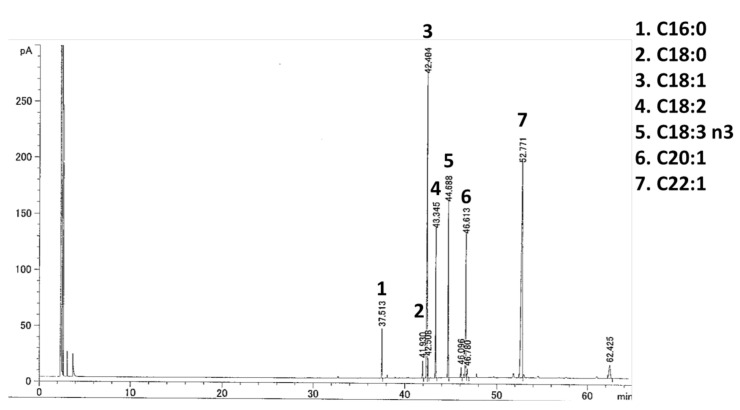
Gas chromatograph with flame ionization detector (GC-FID) chromatogram of YMO.

**Figure 3 molecules-26-00546-f003:**
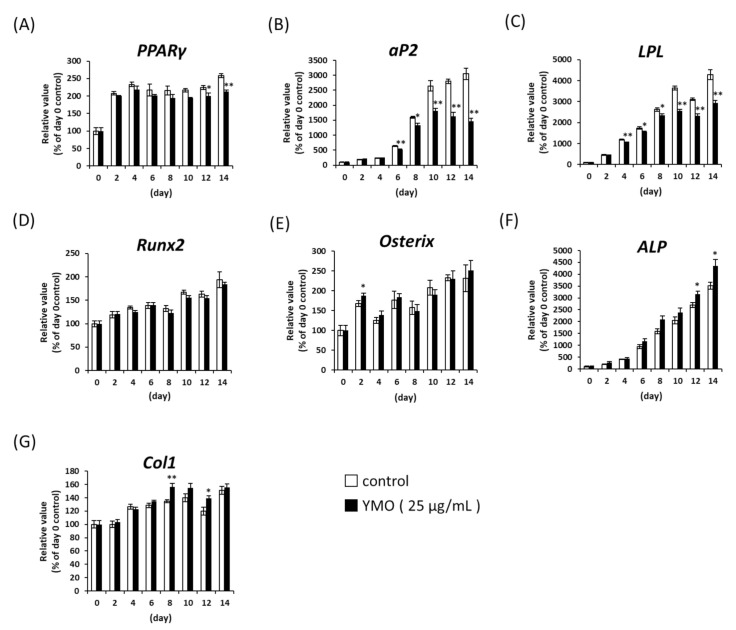
Effect of YMO on the expression of adipocyte or osteoblast marker genes. C3H10T1/2 cells were differentiated and treated with ethanol or YMO (25 µg/mL) for 14 days. mRNA expression levels of adipocyte marker genes (**A**–**C**) and osteoblast marker genes (**D**–**G**) were measured by real-time PCR. Relative mRNA expressions were expressed as percentage of day 0 control. Values are mean ± SE, *n* = 4–5. *** p* < 0.01, ** p* < 0.05 versus each day control.

**Figure 4 molecules-26-00546-f004:**
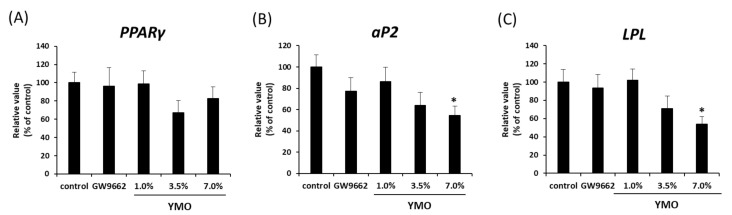
Effect of YMO on the expression of adipocyte differentiation marker genes in perirenal adipose tissue in KK-A^y^ mice. Perirenal adipose tissue was collected from KK-A^y^ mice fed with 1.0, 3.5, or 7.0% YMO for 16 weeks. mRNA expression levels of *PPARγ* (**A**), *aP2* (**B**), and *LPL* (**C**) were measured by real-time PCR. Relative mRNA expressions were expressed as percentage of control. Values are mean ± SE, *n* = 10–11. ** p* < 0.05 versus control.

**Figure 5 molecules-26-00546-f005:**
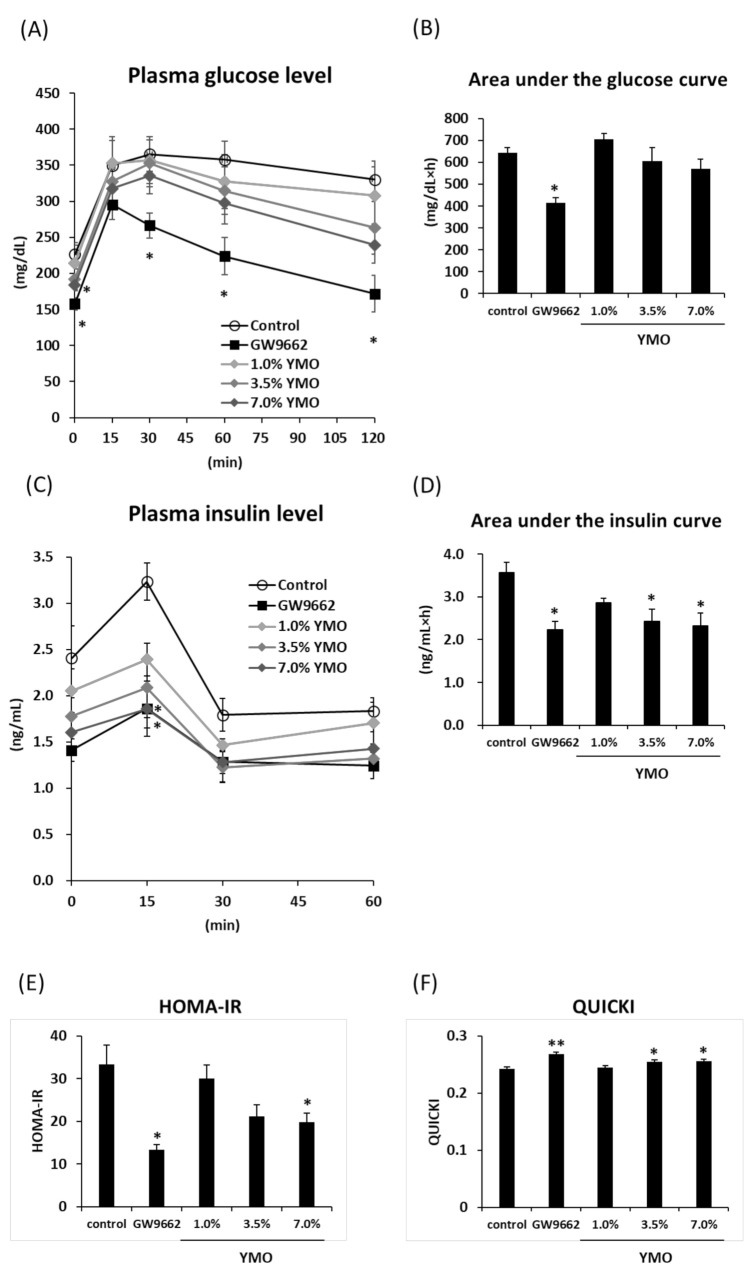
Effect of YMO on plasma glucose and insulin levels during oral glucose tolerance test (OGTT) in KK-A^y^ mice. OGTT was performed at 15 week of animal experiment. Plasma glucose level (**A**), area under the glucose curve (**B**), plasma insulin level (**C**), area under the insulin curve (**D**), homeostasis model assessment-insulin resistance (HOMA-IR) (**E**), quantitative insulin-sensitivity check index (QUICKI) (**F**) were measured in KK-A^y^ mice fed with 1.0, 3.5, or 7.0% YMO. Values are mean ± SE, *n* = 7–10 (**A**,**B**) and *n* = 5–7 (**C**–**F**). ** *p* < 0.01, * *p* < 0.05 versus control.

**Figure 6 molecules-26-00546-f006:**
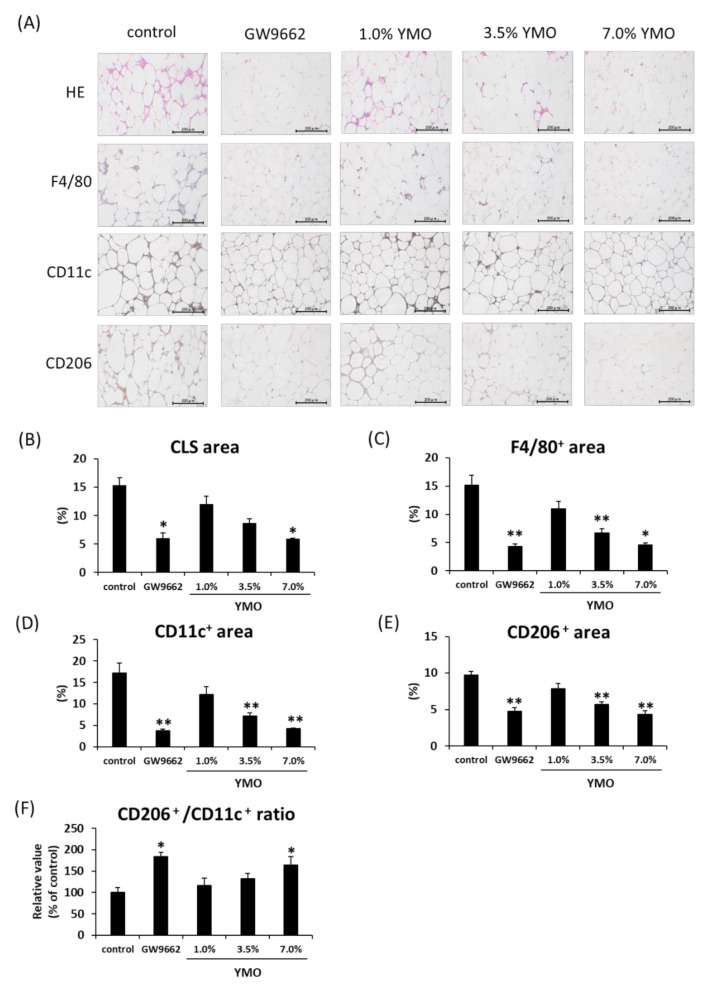
Effect of YMO on macrophage infiltration into perirenal adipose tissue in KK-A^y^ mice. Perirenal adipose tissue was collected from KK-A^y^ mice fed with 1.0, 3.5, or 7.0% YMO for 16 weeks. Paraffin sections of adipose tissue were stained with hematoxylin & eosin (HE), or immunostained with anti-F4/80, CD11c, or CD206 antibody. (**A**) Histological analytic image of HE staining and immunostainings, (**B**) Area of crown like structure (CLS), (**C**) F4/80^+^ area, (**D**) CD11c^+^ area, (**E**) CD206^+^ area, (**F**) Ratio of CD206^+^ area/CD11c^+^ area. Values are mean ± SE, *n* = 7. ** *p* < 0.01, * *p* < 0.05 versus control.

**Figure 7 molecules-26-00546-f007:**
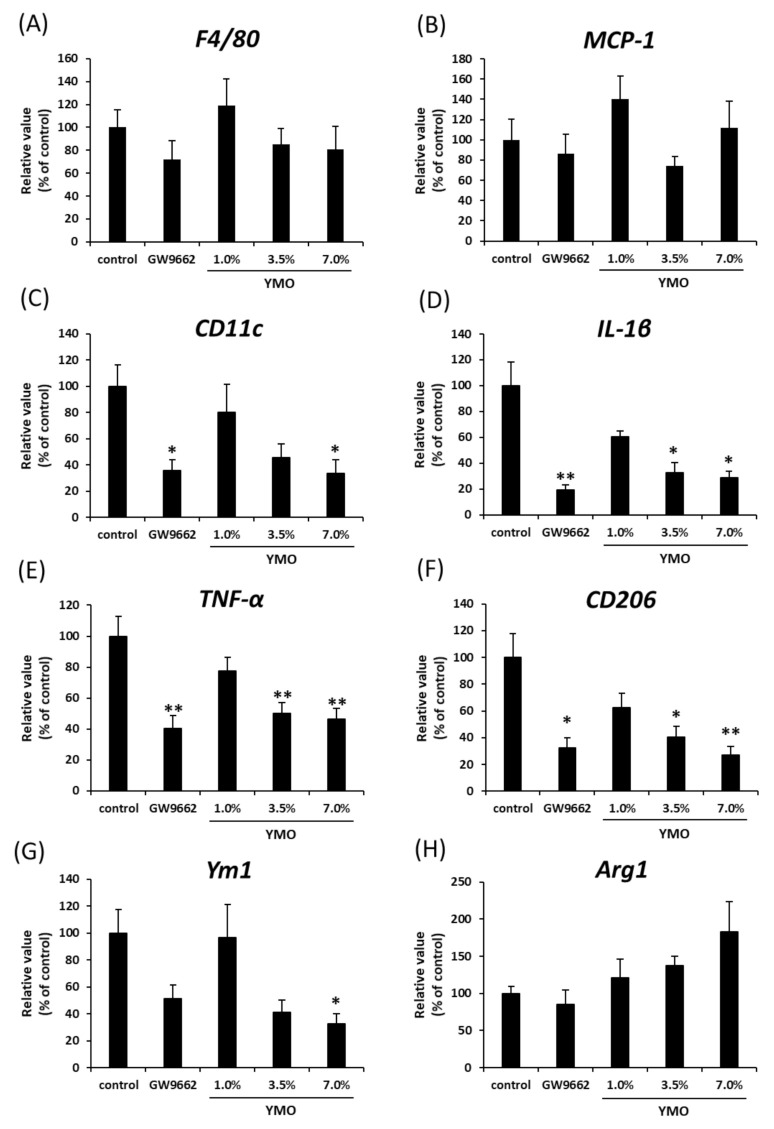
Effect of YMO on the expression of macrophage marker genes in perirenal adipose tissue in KK-A^y^ mice. mRNA expression levels of F4/80 (**A**), MCP-1 (**B**), M1 macrophage markers (**C**–**E**), and M2 macrophage markers (**F**–**H**) in perirenal adipose tissue were measured as described in Figure 3 legend. Values are mean ± SE, *n* = 10–11. ** *p* < 0.01, * *p* < 0.05 versus control.

**Figure 8 molecules-26-00546-f008:**
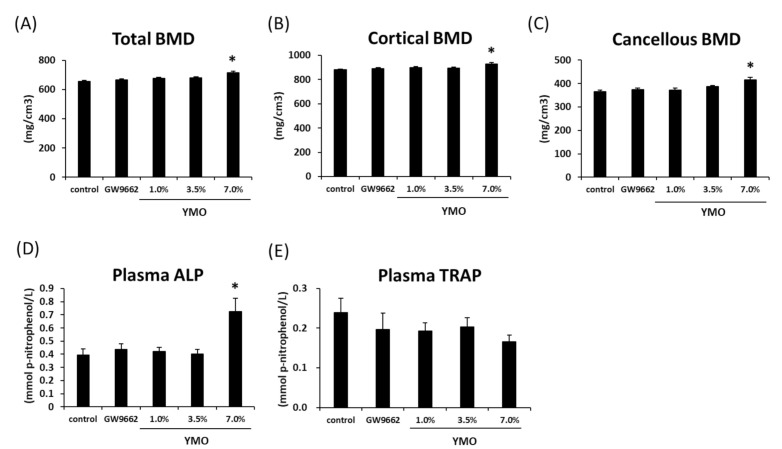
Effect of YMO on bone mineral density (BMD) and plasma bone metabolic markers in KK-A^y^ mice. Right femur was collected from KK-A^y^ mice fed with 1.0, 3.5, or 7.0% YMO for 16 weeks. Total BMD (**A**), cortical BMD (**B**), and cancellous BMD (**C**) were measured by peripheral quantitative computed tomography (pQCT). Plasma ALP (**D**) and TRAP (**E**) activities were measured as described in “Materials and Methods”. Values are mean ± SE, *n* = 10–11. ** p* < 0.05 versus control.

**Table 1 molecules-26-00546-t001:** Composition of fatty acids in YMO.

Fatty Acid (%)	YMO
C16:0	2.61
C18:0	1.05
C18:1	24.35
C18:2	8.92
C18:3 n3	10.49
C20:1	10.88
C22:1	36.97
Others	4.73

**Table 2 molecules-26-00546-t002:** Effect of YMO on final body weight and white adipose tissue weight in KK-A^y^ mice.

	Control	GW9662	1.0% YMO	3.5% YMO	7.0% YMO
Final body weight (g)	42.29 ± 0.70	40.16 ± 0.95	41.98 ± 1.36	39.59 ± 0.68 *	41.41 ± 0.84
Subcutaneous adipose tissue (g)	1.62 ± 0.08	1.41 ± 0.05	1.63 ± 0.10	1.21 ± 0.10 *	1.47 ± 0.07
Perirenal adipose tissue (g)	0.91 ± 0.09	0.94 ± 0.07	0.88 ± 0.10	0.66 ± 0.06	0.59 ± 0.08 *
Mesenteric adipose tissue (g)	1.09 ± 0.04	0.99 ± 0.05	1.21 ± 0.03	0.96 ± 0.06	1.12 ± 0.05
Epididymal adipose tissue (g)	1.04 ± 0.04	0.98 ± 0.03	1.03 ± 0.03	0.96 ± 0.04	1.05 ± 0.06

Values are mean ± SE, *n* = 10–11. * *p* < 0.05 versus control.

**Table 3 molecules-26-00546-t003:** Primers for real-time PCR.

*Genes*	Accession Number	Forward (5′→3′)	Reverse (5′→3′)
*PPARγ*	NM_001127330	GGAGATCTCCAGTGATATCGACCA	ACGGCTTCTACGGATCGAAACT
*aP2*	NM_001122948	AAGACAGCTCCTCCTCGAAGGTT	TGACCAAATCCCCATTTACGC
*LPL*	NM_008509	GCCCAGCAACATTATCCAGT	GGTCAGACTTCCTGCTACGC
*Runx2*	DQ065175	CCCAGCCACTTTACCTACA	TATGGAGTGCTGCTGGTCTG
*Osterix*	AF184902	ACTCATCCCTAATGGCTCGTG	GGTAGGGAGCTGGGTTAAGG
*ALP*	BC065175	GCTGATCATTCCCACGTTTT	CTGGGCCTGGTAGTTGTTGT
*Col1*	NM_007742.4	GAGCGGAGAGTACTGGATCG	GCTTCTTTTCCTTGGGGTTC
*F4/80*	X93328.1	TTTCCTCGCCTGCTTCTTC	CCCCGTCTCTGTATTCAACC
*MCP-1*	NM_011333.3	AGGTCCCTGTCATGCTTCTG	TCTGGACCCATTCCTTCTTG
*CD11c*	NM_007482	TGGGTTTGTTTCTTGTCTTG	GCCTGTGTGATAGCCACATTT
*IL-1β*	NM_008361.4	GCCCATCCTCTGTGACTCAT	AGGCCACAGGTATTTTGTCG
*TNF-α*	NM_013693	ACACTCAGATCATCTTCTCAAAATTCG	GTGTGGGTGAGGAGCACGTAGT
*CD206*	NM_008625	GCGCTGCGTGGACGCTCTAA	ACAGGGTGACGGAAGCCCAGT
*Ym1*	NM_009892	AGAAGGGAGTTTCAAACCTGGT	GTCTTGCTCATGTGTGTAAGTGA
*Arg1*	NM_007482	CAGTTGGAAGCATCTCTGGC	GTGAGCATCCACCCAAATGAC
*36B4*	NM_007475.5	TGTGTGTCTGCAGARCGGGTAC	CTTTGGCGGGATTAGTCGAAG

## Data Availability

Data is contained within the article or Appendix A.

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
