# Peer review of "Erucic Acid-Rich Yellow Mustard Oil Improves Insulin Resistance in KK-Ay Mice"

_molecules, 2021, doi:10.3390/molecules26030546_

Round 1

Reviewer 1 Report

The manuscript titled as ‟Erucic acid-rich yellow mustard oil improves insulin resistance in KK-Ay mice” is well designed. Graphics and tables are clearly presented. Authors have also evaluated YMO intake significantly decreased the surrogate indexes for insulin resistance and the infiltration of macrophages into adipose tissue as well as can ameliorate obesity-induced metabolic disorders. In conclusion, the manuscript can be accepted to be published in Molecules.

Reviewer 2 Report

  1. Erucic acid seems to be activator PPARγ as mentioned by Johnson (1997). It seems erucic acid activates the PPARγ, which leads to reduction of transcriptional activity PPARγ as a compensatory mechanism. Would you please explain this? Structural requirements and cell-type specificity for ligand activation of peroxisome proliferator-activated receptors. T E Johnson 1, M K Holloway, R Vogel, S J Rutledge, J J Perkins, G A Rodan, A Schmidt. J Steroid Biochem Mol Biol. Sep-Oct 1997; 63 (1-3): 1-8.
  2. There was no significant changes in Runx2 and Osterix (figure 2), then making comments about affecting osteoblast differentiation is not acceptable.
  3. The changes in bone mineral density is not remarkable too.

Reviewer 3 Report

Manuscript ID: 

Title: Erucic acid-rich yellow mustard oil improves insulin resistance in KK-Ay mice

The authors investigated the effects of erucic acid-rich yellow mustard oil (YMO) on obese/diabetic KK-Ay mice. To know the YMO compositions, it has been conducted by the hydrolysis and derivatization, and analyzed by GC/MS. In general, the authors have completed a reasonable study with very informative data on the YMO compositions and bioactivities. However, some of minor concerns would be suggested and requested for further improvement in the manuscript.

  1. Line 21

Please change the unit of dose of 7.0% by using mg/Kg or, g/Kg.

  1. GC/MS analysis
    The GC/MS chromatogram of YMO should be presented for relating to the data of Table 1.

    3. Line 219
    The meaning in the investigation of CD11C should be addressed in the Discussion. Similar to the others too, if they were necessarily.

  1. Finally, please explain that diet treated in the control group would induce an insulin resistance symptom.
